# Orbital-symmetry effects on magnetic exchange in open-shell nanographenes

Qingyang Du[1], Xuelei Su [1], Yufeng Liu[2], Yashi Jiang[2], Can Li [2], KaKing Yan [1], Ricardo Ortiz[3] ✉, Thomas Frederiksen [3,4] ✉, Shiyong Wang [2,5] ✉ & Ping Yu [1] ✉

Open-shell nanographenes appear as promising candidates for future applications in spintronics and quantum technologies. A critical aspect to realize this potential is to design and control the magnetic exchange. Here, we reveal the effects of frontier orbital symmetries on the magnetic coupling in diradical nanographenes through scanning probe microscope measurements and different levels of theoretical calculations. In these open-shell nanographenes, the exchange energy exhibits a remarkable variation between 20 and 160 meV. Theoretical calculations reveal that frontier orbital symmetries play a key role in affecting the magnetic coupling on such a large scale. Moreover, a triradical nanographene is demonstrated for investigating the magnetic interaction among three unpaired electrons with unequal magnetic exchange, in agreement with Heisenberg spin model calculations. Our results provide insights into both theoretical design and experimental realization of nanographene materials with different exchange interactions through tuning the orbital symmetry, potentially useful for realizing magnetically operable graphene-based nanomaterials.

Magnetism in nanographenes has attracted a lot of attention due to its unique properties such as weak spin-orbit coupling and long spin coherence time[1,2], compared with conventional magnetism originating from d- or f-block elements. With the development of on-surface synthesis[3–9], multiradical nanographenes can be precisely fabricated with well-defined π-electron topologies via the design of molecular precursors[10–13], which are ideal platforms for investigating the underlying mechanism of magnetic-exchange interactions[14]. To induce magnetism in nanographenes, embedding substitutional heteroatoms[15–18] or incorporating pentagon rings[19–23] have been employed in nanographenes using on-surface synthesis. In addition to these approaches, sublattice imbalance in a bipartite lattice can also generate net spins as predicted by the Ovchinnikov's rule and Lieb's theorem[24,25]. Accordingly, triangulene and its π-extended homologs

have been fabricated[26–28], which show high-spin ground states in agreement with theoretical predictions. For realizing spin-logic devices or molecular switches at room temperature[29,30], robust magnetic ordering with large magnetic coupling strength is needed. Towards this goal, the largest magnetic-exchange coupling has been reported in rhombus-shaped nanographenes with zigzag periphery (rhombene)[31,32].

Besides the interest in the magnetic properties of triangulenes and rhombenes, they can also serve as building blocks for constructing spin networks with collective quantum behaviors to explore quantum phases of matter[33–40]. The linkers that connect adjacent nanographene units have played a key role in engineering the magnetic coupling among building blocks. As an example[12,34,41–43], antiferromagnetism can be tuned to ferromagnetism by adjusting the C-C bonding sites from

[1]School of Physical Science and Technology, ShanghaiTech University, 201210 Shanghai, China. [2]Key Laboratory of Artificial Structures and Quantum Control (Ministry of Education), Shenyang National Laboratory for Materials Science, School of Physics and Astronomy, Shanghai Jiao Tong University, Shanghai 200240, China. [3]Donostia International Physics Center (DIPC) – UPV/EHU, 20018 San Sebastián, Spain. [4]IKERBASQUE, Basque Foundation for Science, 48013 Bilbao, Spain. [5]Tsung-Dao Lee Institute, Shanghai Jiao Tong University, Shanghai 200240, China. ✉e-mail: roc6493@gmail.com; thomas_frederiksen@ehu.eus; shiyong.wang@sjtu.edu.cn; yuping@shanghaitech.edu.cn

the same to opposite sublattices. The strength of the magnetic-exchange interaction can also be manipulated by including a connection spacer[44] or incorporating a pentagon ring that affects the orbital-density overlap at the connecting region[36]. However, the effects of orbital symmetry on the magnetic-exchange interactions have been rarely addressed, whilst it is known to be critical for some chemical and physical processes. For example, a cycloaddition reaction requires the symmetry of the highest occupied molecular orbital (HOMO) to be the same as the symmetry of the lowest unoccupied molecular orbital (LUMO)[45]. More recently, it was fabricated a $\pi$-conjugated polymer of acenes that shows a quantum topological phase transition with an avoided level crossing of HOMO and LUMO[46]. Considering finite multiradical nanographenes, frontier orbitals consist of hybridized unpaired electrons, hence one may anticipate that their symmetry should affect the magnetic properties of the molecule[47–49].

A previous work by one of us demonstrated that molecular symmetries are crucial for determining the exchange coupling of diradical nanographenes[50]. Inspired by this, we investigated the effects of the parity symmetry of frontier orbitals on the magnetic-exchange of open-shell nanographenes. We employ phenalenyl as the building block to fabricate various nanographene trimers with different conjugation symmetries by using an on-surface synthesis approach. In experiments, all three phenalenyl groups can be passivated with additional hydrogen atoms on the radical sites. Through tip-induced dehydrogenation at different passivation sites, three types of diradicals, that we call D1, D2 and D3, can be obtained, as shown in Fig. 1a. According to Lieb's theorem[25], D3 has a sublattice imbalance and thus hosts a net spin of $S = 1$. For the other diradicals, although they have balanced sublattices, they may also exhibit open-shell character by comparing their Kekulé with non-Kekulé structures, whose energy cost of having unpaired electrons could be

compensated by the extra Clar sextets (Fig. 1b, c)[51]. Using a combination of scanning tunneling spectroscopy (STS) and inelastic electron tunneling spectroscopy (IETS), we find that D1 and D2 indeed show local magnetic moments with antiferromagnetic coupling. Their exchange interactions are 20 and 160 meV respectively, showing a factor of 8 variation. Our theoretical calculations suggest that this huge difference is due to their different frontier orbital symmetries. We found that distant-neighbor hopping also significantly affects the magnetic-exchange by including the third-nearest-neighbor hopping in the calculations. Moreover, we reveal the magnetic ground state and excited states of a triradical phenalenyl trimer (T) with unequal magnetic coupling strengths, consistent with Heisenberg spin model calculations. Our results demonstrate a new approach for effectively varying the exchange interaction strength in a nanographene through engineering its frontier orbitals symmetry, which is inspiring for both theoretical design and practical realization of spintronic devices.

## Results

### Nanographene diradicals and a triradical obtained by on-surface synthesis and atomic manipulation

To fabricate the previously introduced open-shell nanographenes, precursor 1 in Fig. 2a is designed as an 8-methyl-naphthalene and a benzene ring with two bromide substitutes. Precursor 1 was first deposited on Au(111) with submonolayer coverage and then annealed to the temperatures of 433 K and 523 K subsequently for triggering debrominative cycloaddition and cyclodehydrogenation reactions. As the reaction scheme shows in Fig. 2a, due to the adsorption handedness, the [2 + 2 + 2] cycloaddition reaction[52,53] of three precursors 1 with the same or opposite adsorption handedness will lead to different products 2 and 3 formed by three phenalenyl units conjugated by the additional formed phenyl ring in different orientations.

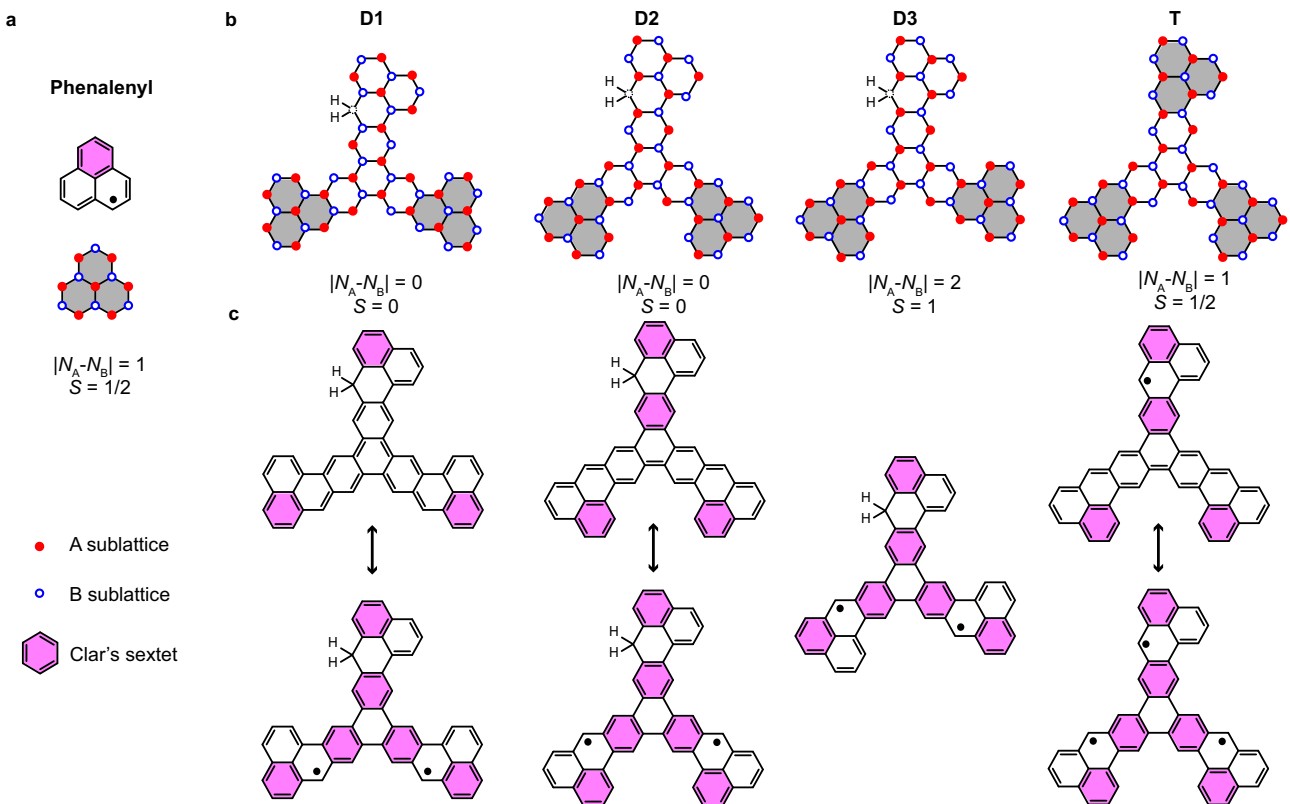

**Fig. 1 | Engineering magnetic-exchange interaction of diradical phenalenyl-based nanographenes through different $\pi$-conjugation symmetry. a** Chemical structures of phenalenyl and its corresponding Clar formula. **b** Three types of nanographene diradicals and a triradical labeled as D1, D2, D3 and T, respectively. **c** Chemical sketch of possible Kekulé and non-Kekulé resonance structures for D1-D3 and T.

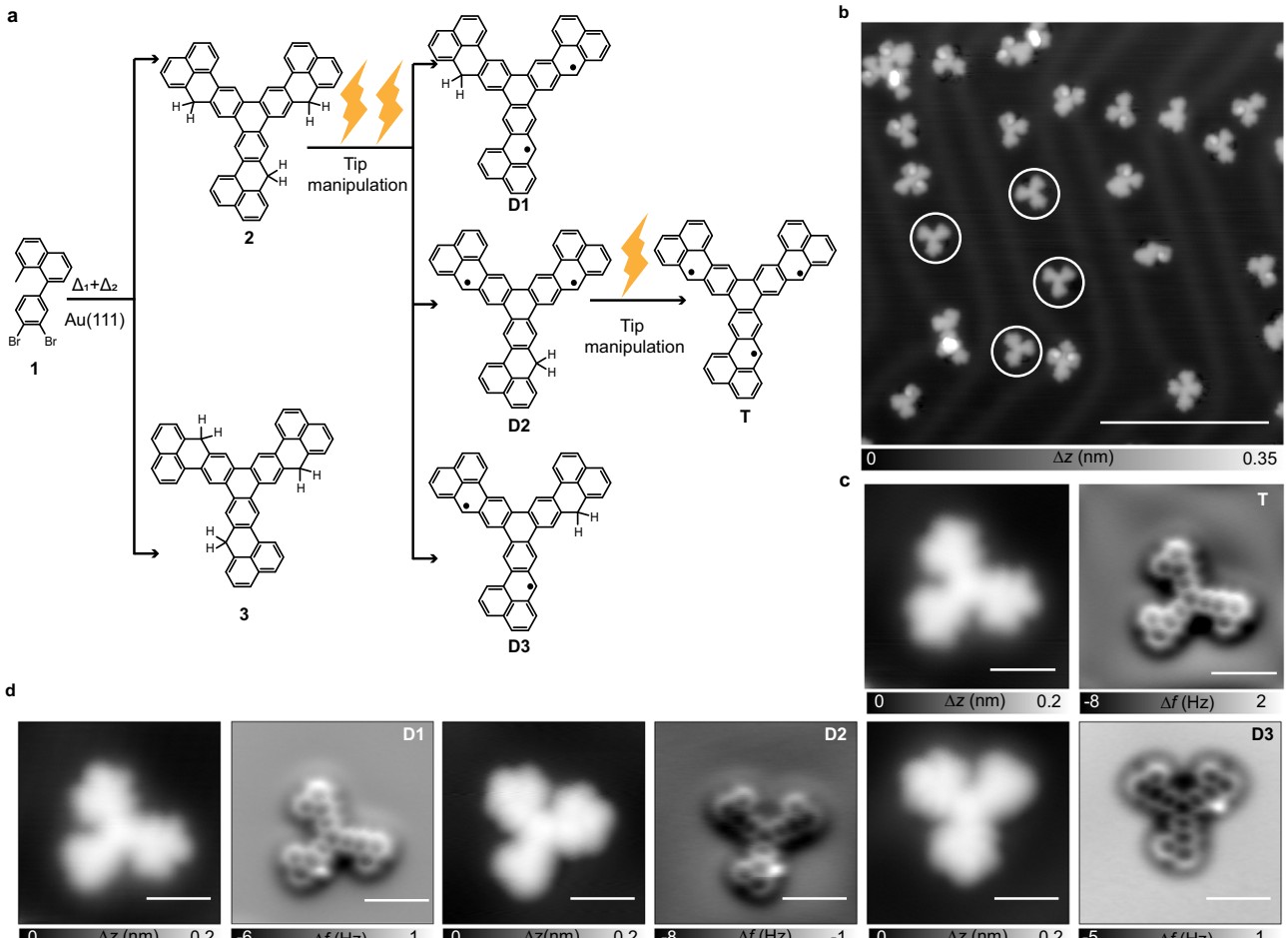

**Fig. 2 | On-surface synthesis and structural characterization of diradicals D1-D3 and triradical T. a** Synthetic route towards **D1-D3** and **T** through stepwise annealing and tip-induced dehydrogenation. **b** Overview STM image ($V = 300$ mV, $I = 50$ pA) after annealing precursor **1** on Au(111), revealing individual molecules. **c** Zoom-in STM image ($V = 300$ mV, $I = 100$ pA) and corresponding bond-resolved AFM image of **T**. **d** Zoom-in STM images ($V = 300$ mV, $I = 100$ pA) and the bond-resolved AFM images of **D1-D3**. Scale bars: 10 nm in (**b**), 1 nm in others.

Figure 2b shows the overview image of the main products **2** and **3** by using scanning tunneling microscopy (STM), most of which are passivated by two hydrogen atoms on the phenalenyl units during the on-surface synthesis process[12,42]. Upon tip manipulation, the desired diradicals (**D1, D2, D3**), as well as the triradical (**T**), can be obtained by a voltage pulse at different passivation sites. Their chemical structures are further characterized by bond-resolved atomic force microscopy (AFM) measurements[54,55], which are shown sequentially in Fig. 2c, d. In the AFM images of **D1, D2** and **D3**, the remaining hydrogen passivation site can be observed due to the bright contrast from the additional hydrogen atom repulsion.

## Huge difference of magnetic-exchange interactions in diradicals D1 and D2 measured by spin-flip spectroscopy

The AFM images and the corresponding chemical structures of **D1-D3** are shown in Fig. 3a. One of the three phenalenyl units is passivated by two hydrogen atoms, thus having two unpaired electrons in the π system. To investigate their magnetic properties, d$I$/d$V$ spectroscopy is measured on different positions of **D1-D3** as marked in the AFM images (Fig. 3b)[56]. For **D1**, symmetric steps around the Fermi level are detected at ±20 meV, indicating an inelastic excitation[11,22]. We attribute these to spin-flip processes corresponding to the singlet-triplet spin excitation, corresponding to an antiferromagnetic coupling of 20 meV. In sharp contrast, the corresponding d$I$/d$V$ spectra taken on **D2** exhibit a substantially increased excitation threshold of 160 meV. Additionally, a more pronounced asymmetric line shape of d$I$/d$V$

spectra is observed on **D2** compared to that of **D1**, since **D2** has a high-spin-excitation energy close to its SOMO resonate state, thus leading to a pronounced height in the negative bias[32,57]. These results demonstrate that the exchange interaction strength can have a variation of about one order-of magnitude just by changing the conjugation symmetry. (The corresponding d$^2I$/d$V^2$ spectra can be found in Supplementary Fig. 16). Regarding **D3**, as predicted by Lieb's theorem in Fig. 1b, it should have an open-shell triplet ground state, which was confirmed by a Kondo peak feature at the Fermi level in the d$I$/d$V$ spectra (Fig. 3b)[12,42]. To visualize the spatial distribution of the spin excitation and the Kondo effect, d$I$/d$V$ maps recorded at the excitation energies of ±20 meV, ±160 meV, and the Fermi energy are shown in Fig. 3c, respectively. They all agree well with the simulated STM images using the singly- occupied orbitals from Fig. 3c, suggesting that the studied diradical nanographenes have an open-shell ground state. Moreover, the large range of the d$I$/d$V$ spectra measured on **D1** and **D2** (Supplementary Fig. 17), in which the singly-occupied and singly-unoccupied molecular orbitals (SOMOs/SUMOs) are detected, further confirm their magnetic ground states.

## Orbital symmetry effects on the exchange interaction investigated by theoretical calculations

The magnetic ground states of the diradicals have been addressed by calculations at different levels. Since **D3** hosts a high-spin ground state, that is well understood by theory[25] and experiments[42,43] due to its sublattice imbalance, we focus on the competing cases of **D1** and **D2**

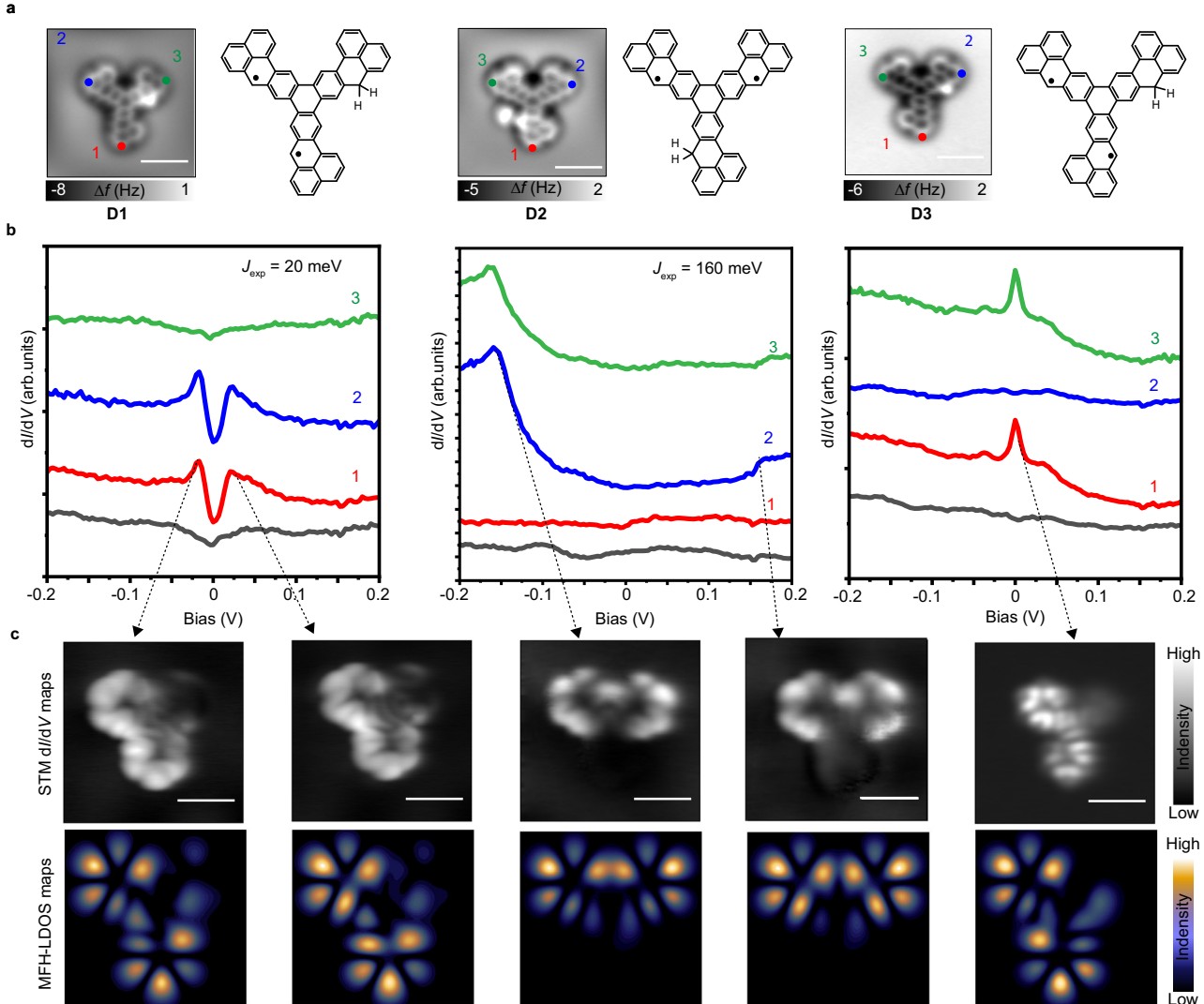

**Fig. 3 | Electronic and magnetic characterization of D1-D3. a** The AFM images and corresponding molecular models for **D1-D3. b** d$I$/d$V$ spectra acquired at different positions marked with filled circles in (**a**), respectively ($V = 200$ mV, $I = 300$ pA, $V_{rms} = 5$ mV). **c** Constant current d$I$/d$V$ maps of spin excitations ($I = 100$-500 pA, $V_{rms} = 5$ mV) and the constant-height Kondo map ($V_{rms} = 2$ mV) with the corresponding theoretical simulation maps shown below. Scale bars : 1 nm.

(Fig. 4). According to simple, but illustrative mean-field Hubbard (MFH) calculations (Supplementary Table 1), both cases have an open-shell $S = 0$ ground state and an $S = 1$ first excited state, suggesting that the energy cost of introducing two unpaired electrons is smaller than the energy gain of adding two Clar sextets (cf. Figure 1b). To calculate the excitation energy of the diradicals and address the enhanced electron-electron interactions in such small nanographenes, complete active space Hubbard (CAS-Hubbard) calculations[58] and a robust quantum chemistry method (CASSCF-NEVPT2) were performed[59-62]. The spin density distributions, calculated with MFH, are shown in Fig. 4a. Although **D1** and **D2** share the same (singlet) ground and the first excited (triplet) magnetic states, the strength of their exchange coupling varies substantially according to CAS-Hubbard and CASSCF-NEVPT2 results (Fig. 4b), which is consistent with the experimental evidence that **D2** has a much larger exchange coupling than **D1**. The CASSCF-NEVPT2 method appears to overestimate the spin-excitation energy in **D2**, which can be attributed to a further renormalization as consequence of the surface. On the other hand, the CAS-Hubbard method (only considering the nearest-neighbor hopping term) clearly overestimates (underestimates) the spin-excitation energy of **D1** (**D2**) for typical values of $U$ ($|t| < U < 2.2|t|$). To rationalize the experimental results, we find it necessary to add a third-nearest-neighbor hopping

term in CAS-Hubbard ($t_3 = -0.4$ eV), suggesting that distant-neighbor hopping processes may play an important role in the exchange coupling in **D1** and **D2**.

Figure 5 depicts the physical nature behind such large variations. To understand this mechanism, the MFH frontier orbital wave functions of molecules **D1** and **D2** are calculated without electron-electron interactions ($U = 0$). The HOMO and LUMO are shown in Fig. 5a, where the middle dashed lines represent the mirror plane normal to the $\pi$ system. The HOMO of **D1** is antisymmetric with respect to the mirror plane, while the HOMO of **D2** is symmetric. We notice that the different frontier orbital symmetry changes the orbital-density overlap at the linkers, and thus modifies the exchange coupling. As highlighted in Fig. 5b, the HOMO wave function of **D1** is antisymmetric with respect to the mirror plane and propagates in the upper branch of the molecular backbone. Thus, the singly-occupied orbital-density per site at the connecting region (dashed box) of **D1** is effectively reduced, resulting in a smaller exchange interaction strength for **D1**. In contrast, **D2** has a symmetric HOMO, wave function that does not propagate in the upper branch of the molecular backbone. Therefore, the singly-occupied orbital-density distribution is not reduced at the connecting region for **D2**. As a consequence, **D1** has a smaller coupling strength than

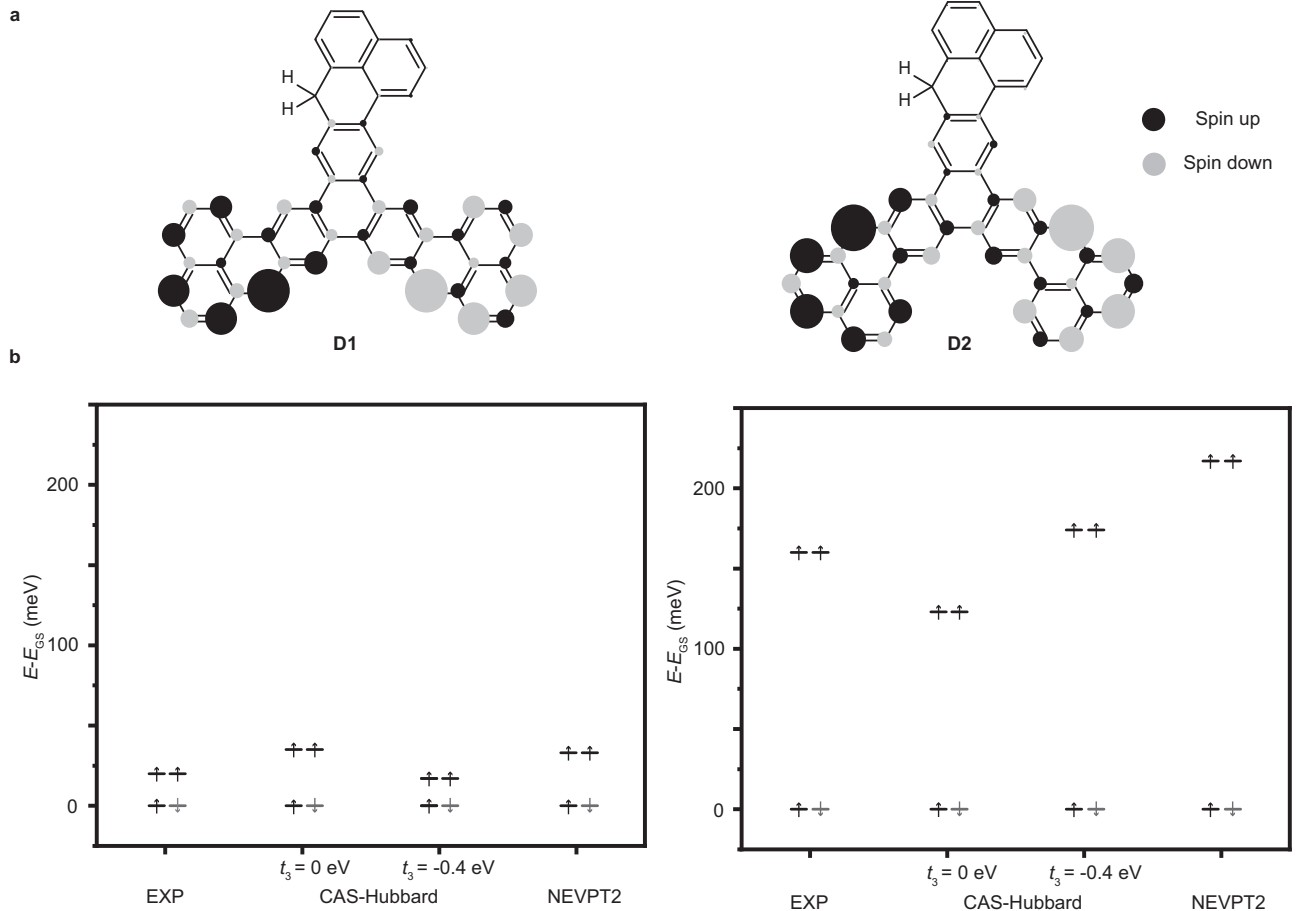

**Fig. 4 | Calculated spin density and excitation energy for D1 and D2 with different theoretical methods. a** Spin density distributions of **D1/D2** with MFH calculations ($U = 3.5$ eV). Black/gray denote spin up/down density. **b** Energies of open-shell triplet states with respect to the open-shell singlet ground state (GS) of **D1** and **D2** calculated with CASSCF-NEVPT2 and CAS-Hubbard methods with different $t_3$ values ($U = 4.05$ eV).

**D2** due to the reduced orbital-density overlap at the connecting region.

Orbital symmetry does not only affect the orbital-density distributions at the connections, but also the effective coupling through distant-neighbor hopping. As shown in Fig. 4b, although the calculated exchange interactions of **D1** is smaller than that of **D2** only considering the $t_1$ hopping term, the exchange interaction of **D1** and **D2** obtained by CAS-Hubbard is remarkably different from the experimental values. To quantitatively explain the experiments, we find that including a $t_3$ hopping term causes the exchange interaction to decrease for **D1**, while increases for **D2**, now in better agreement with the experimental values. These results suggest that the $t_3$ term also has an impact on the exchange interaction, that is included in the CASSCF-NEVPT2 calculation. (The next-nearest hopping $t_2$ of the honeycomb lattice is less important in our case, cf. Supplementary Fig. 19). This mechanism is illustrated by the linear combinations $\psi_{L,R} = \frac{1}{\sqrt{2}}(\psi_{HOMO} \pm \psi_{LUMO})$ as shown in Fig. 5c, where in all cases $\psi_{L,R}$ are spatially separated, and localized at the left or right side of the diradical. The $t_3$ hopping effect on the coupling strength can be captured by considering the effective hopping $t_3$ between $\psi_L$ and $\psi_R$, defined as $\langle\psi_L|\hat{H}_{t3}|\psi_R\rangle$. At the connecting region, the orbital sign at two sites connected by $t_3$ is the same for **D1** with antisymmetric HOMO, but opposite for **D2** with symmetric HOMO (as marked by red/blue arrows in Fig. 5c). The change of HOMO-LUMO energy gap depends on $\langle\psi_L|\hat{H}_{t3}|\psi_R\rangle$, which is negative for **D1** and positive for **D2** considering the value of $t_3$ is negative for graphene systems[63]. As shown in Fig. 5d, the HOMO-LUMO energy gap of **D1** (**D2**) decreases (increases) with the increased magnitude of $t_3$, suggesting a reduced (enhanced) effective coupling strength. As a

result, a difference of about one order-of magnitude of the magnetic exchange in the diradicals can be realized by tuning frontier orbital symmetries as well as the third-nearest-neighbor hopping.

## Electronic and magnetic properties of triradical T

In addition, a triradical can also be studied with our molecules. All extra hydrogen atoms can be dissociated by voltage pulses, thus resulting in a nanographene with three unpaired electrons (Fig. 6a–c). Among these three radical sites, two of them have a ferromagnetic interaction, while the other two pairs have different antiferromagnetic interactions. The unequal coupling compete in this triradical system. d$I$/d$V$ spectra are measured at different sites as marked in Fig. 6a. As we can see in Fig. 6d, a Kondo feature is detected at the Fermi level at position 3, while the spin-flip features with an excitation threshold of 160 meV are observed at positions 1 and 2. The spin-excitation d$I$/d$V$ maps and the Kondo map are shown in Fig. 6e. These results suggest that this triradical nanographene has a ground state of $S = 1/2$. To understand this system, a Heisenberg spin trimer model (Fig. 6f) was solved, the calculation results from Fig. 6g demonstrate that the ground state has $S = 1/2$ and the other two excited states are nearly degenerate with an excitation gap of ~160 meV. CAS-Hubbard calculations have been performed to address spin correlations among these unpaired electrons, showing a good correspondence with the spin model (Fig. 6f). The excitation energy between the $S = 1/2$ ground state and the $S = 3/2$ first excited state are illustrated as a function of $U/|t|$ in Fig. 6h. The excitation energy between $S = 1/2$ and $S = 3/2$ is 170 meV considering $U/|t| = 1.3$ comparable with the Heisenberg

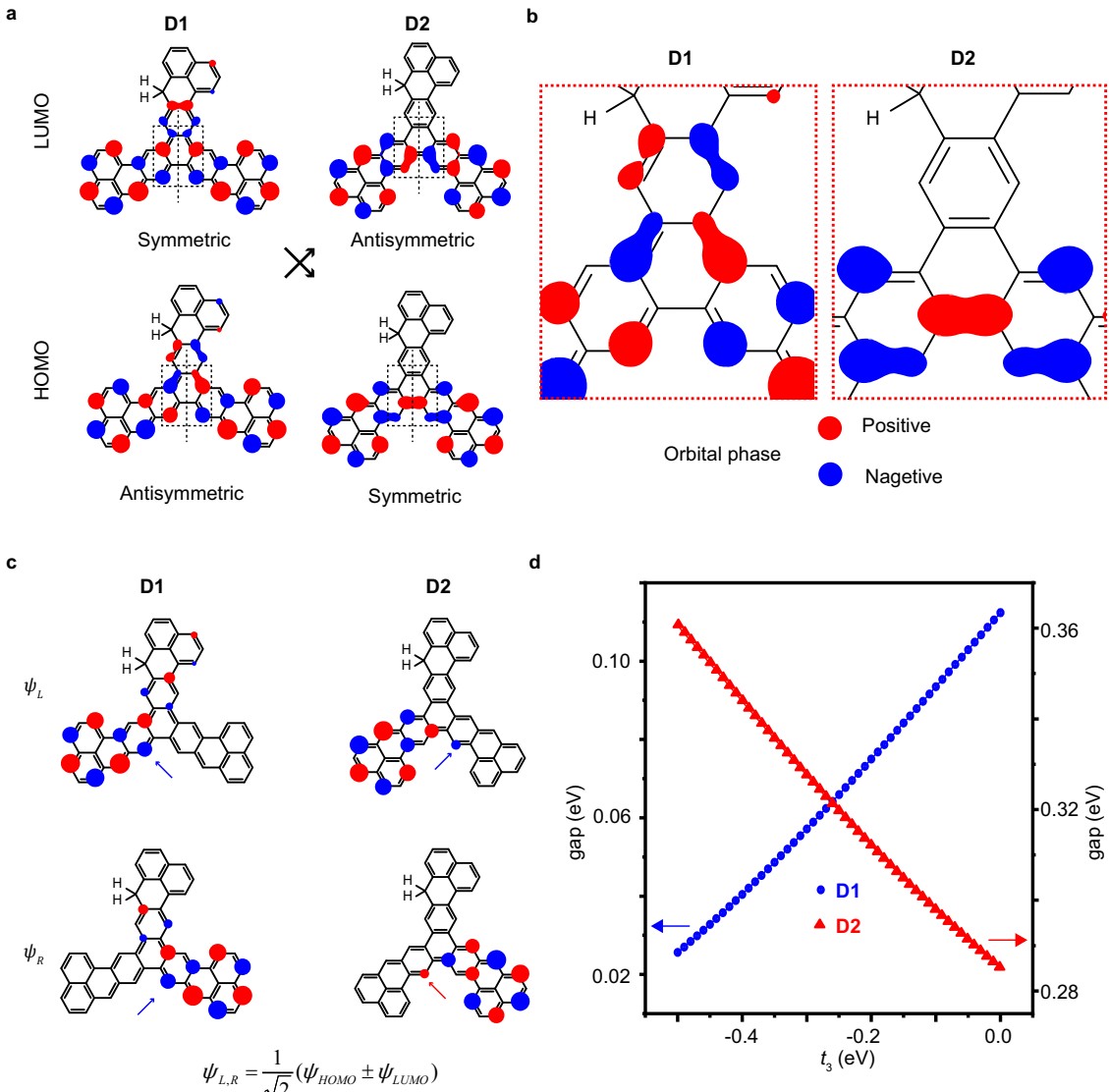

**Fig. 5 | The effect of third-nearest-neighbor hopping on the magnetic coupling tuned by orbital symmetries. a** Symmetry flipping of LUMO/HOMO frontier orbitals between **D1** and **D2** corresponding to the mirror plane as marked by middle dashed lines, the blue and red isosurfaces denote the opposite signs of the orbital phases. **b** HOMO distribution at the conjugation sites dependent on the orbital symmetry. **c** Calculated wave functions $\psi_L$ (top) and $\psi_R$ (bottom) in the form of $\psi_{L,R} = \frac{1}{\sqrt{2}}(\psi_{HOMO} \pm \psi_{LUMO})$. **d** HOMO-LUMO single particle gaps of **D1** and **D2** plotted as a function of the third-nearest hopping term $t_3$ with $U = 0$.

spin model calculations. The spin correlators among the three unpaired electrons are calculated through the term $<\Psi|S_z(i)S_z(j)|\Psi>$, with $\Psi$ being the CAS-Hubbard ground-state wave function and $S_z$ the spin operator. As shown in Fig. 6i, the calculated spin correlator maps show the antiferromagnetic unpaired electrons at the top-right and bottom are strongly correlated, while the one at the top-left is almost uncoupled from the others, in agreement with experiments.

## Discussion

In summary, we have demonstrated an approach for varying the magnetic-exchange interaction in $\pi$-conjugated open-shell nano-graphenes upon tuning the frontier orbital symmetries. Combined with different levels of theoretical calculations and scanning probe microscope measurements, the chemical and electronic structures of various diradical nanographenes have been investigated on surfaces. As the theoretical calculations have demonstrated, their coupling strengths can vary by one order-of-magnitude (20 –160 meV) via flipping the HOMO/LUMO symmetry. Moreover, the competition among

three unpaired electrons with unequal magnetic coupling strengths has been demonstrated, in agreement with Heisenberg spin model calculations. Our results provide insights for obtaining magnetic-exchange interaction in a large-scale through tuning frontier orbital symmetry, which could extend the design strategy for realizing graphene-based spintronic nanomaterials in the future.

## Methods

### Sample preparation and STM/AFM measurements

The STM/AFM experiments for the electronic and chemical structure characterization were performed at $4.7\,K$ with commercial Createc LT-STM/qplus AFM. The Au(111) single-crystal was cleaned by cycles of argon ion sputtering and subsequently annealed to $800\,K$ to get atomically flat terraces. Molecular precursors **1** were thermally deposited on the clean Au(111) surface, and subsequently annealed to $433\,K$ and $523\,K$ to fabricate structure **2** and **3**, The AFM measurements were performed with the qPlus sensor with the resonance frequency of $32.6\,KHz$ and the oscillation amplitude of $50\,pm$. d$I$/d$V$ measurements were performed with an internal lock-in amplifier at frequency of

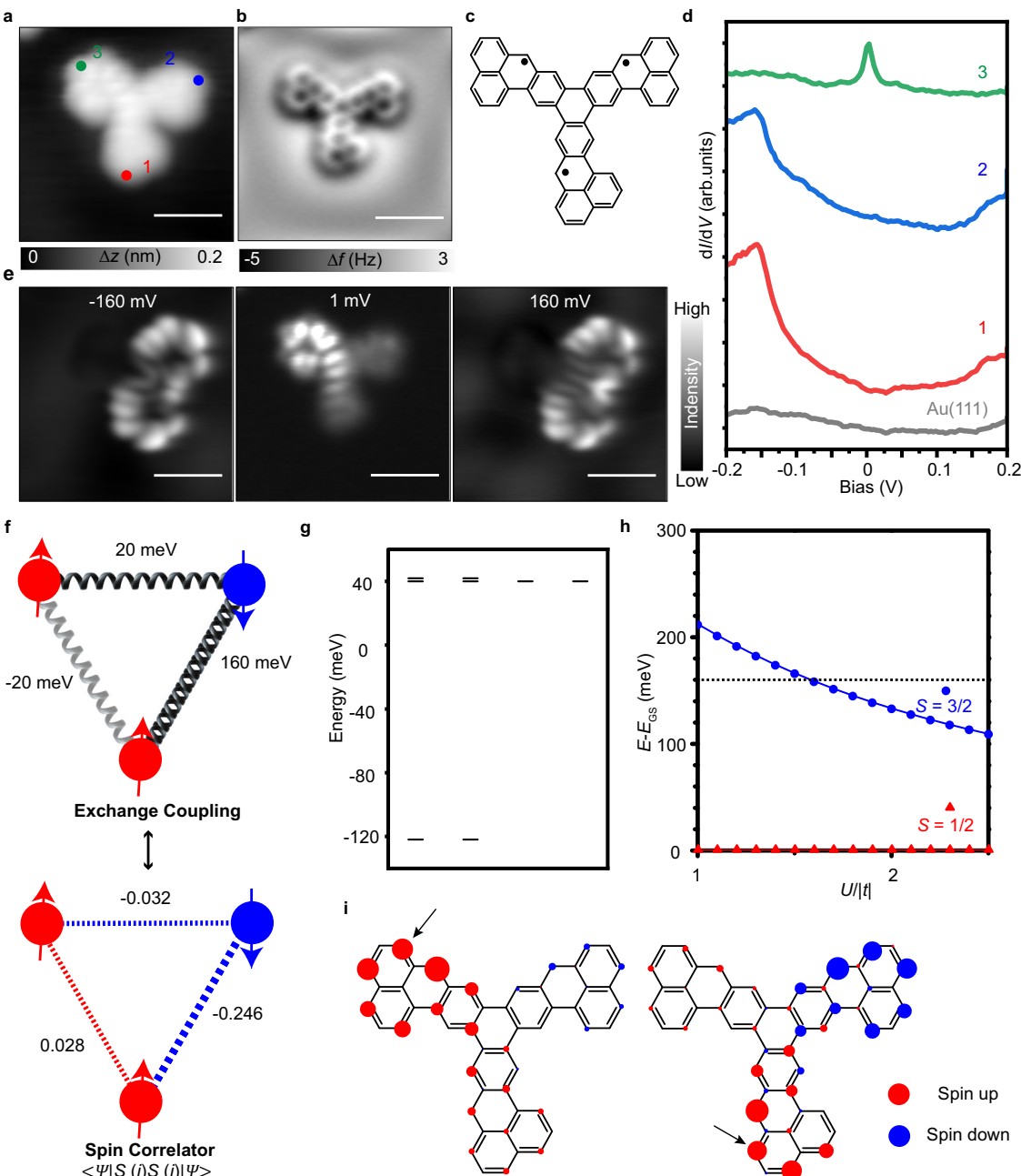

**Fig. 6 | Electronic and magnetic characterization of the triradical T. a, b, c** STM image ($V$ = 300 mV, $I$ = 100 pA), bond-resolved AFM image, and molecular model of **T. d** d$I$/d$V$ spectra acquired at different positions marked with filled circles in (**a**). **e** Constant current d$I$/d$V$ maps acquired at ±160 mV for spin excitations ($I$ = 300-500 pA, $V_{rms}$ = 5 mV) and constant-height d$I$/d$V$ Kondo map ($V_{rms}$ = 2 mV) of **T. f** Schematic spin model with the experimental excitation energies as magnetic exchanges, and spin correlators between three phenalenyl units of **T**, with the values of 0.028, −0.032 and −0.246 calculated for $<\Psi|S_z(i)S_z(j)|\Psi>$. **g** Calculated Heisenberg energy levels of the spin model. **h** Energy difference between first excited state and open-shell ground state (GS) of **T** calculated in CAS (3,3) plotted as a function of $U/|t|$. The black dotted line indicates the experimental gap of 160 meV for **T. i** Spin correlator map calculated by selecting an atom $i$ and running over all the other atoms $j$. The chosen atom $i$ are marked by black arrows. Red/blue denote spin up/down density distribution.

862 Hz. Lock-in modulation voltages for individual measurements were provided in the respective figure caption. All STM/STS and AFM measurements were acquired with CO-functionalized tungsten tip. For the constant-height AFM images, the tip-distance is decreased a few hundred of pm from the STM set point $V$ = 300 mV, $I$ = 50 pA.

**Tight binding (TB) and mean-field Hubbard (MFH) modeling**
The tight-binding (TB) calculations were carried out in the C 2$p_z$-orbital description by numerically solving the Mean-Field-Hubbard

Hamiltonian:

$$\hat{H}_{MFH} = \sum_{k=1}^{3} \sum_{<ij>,\sigma} t_k c_{i,\sigma}^{\dagger} c_{j,\sigma} + U \sum_{i,\sigma} \langle n_{i,\sigma} \rangle n_{i,\bar{\sigma}} - U \sum_{i} \langle n_{i,\uparrow} \rangle \langle n_{i,\downarrow} \rangle \quad (1)$$

with $t_1$, $t_2$, $t_3$ denoting the nearest-neighbor, second nearest-neighbor, third-nearest-neighbor hopping term depending on the bond length between C atoms, and $c_{i,\sigma}^{\dagger}$ and $c_{j,\sigma}$ denoting the spin selective ($\sigma = \{\uparrow, \downarrow\}$) creation and annihilation operators on the

atomic site $i$ and $j$, $U$ the on-site Coulomb repulsion parameter (with $U = 3.5$ eV used here), $n_{i,\sigma}$ the number operator and $\langle n_{i,\sigma} \rangle$ the mean occupation number at site $i$. Numerically solving the model Hamiltonian yields the energy Eigenvalues $E_i$ and the corresponding Eigenstates $\alpha_{ij}$ (amplitude of state $i$ on site $j$) from which the wave functions are computed assuming Slater type atomic orbitals:

$$\psi_i(\vec{r}) = \sum_j a_{i,j} \cdot (z - z_j) \exp(-\zeta|\vec{r} - \vec{r}_j|) \tag{2}$$

with $\zeta = 1.625$ a.u. for the carbon $2p_z$ orbital. The charge density map $\rho(x,y)$ for a given energy range $[\varepsilon_{\min}, \varepsilon_{\max}]$ and height $z_0 = 3.5$ Å is then obtained by summing up the squared wave functions in this chosen energy range:

$$\rho(x,y) = \sum_{i, \varepsilon_i \in [\varepsilon_{\min}, \varepsilon_{\max}]} \psi_i^2(x,y,z_0) \tag{3}$$

Constant charge density maps are taken as a first approximation to compare with experimental STM images.

### Complete active space (CAS) Hubbard and CASSCF-NEVPT2
The CAS-Hubbard calculations are performed as the previous work reported[50]. For **T**, a CAS (3,3) wave function was selected. The energies calculated by the CAS-Hubbard method used the hopping parameters of $t_1 = -2.7$ eV, $t_2 = -0.1$ eV and $t_3 = -0.4$ eV. We also performed ab-initio quantum chemistry calculations (CASSCF), with orbitals calculated with DFT with a PBE functional. We chose CAS(10,10) and 7 roots for **D1** and CAS(12,12) and 6 roots for **D2**, and then corrected with second-order N-electron valence state perturbation theory (NEVPT2).

## Data availability
All data generated in this study are available within the article and supplementary information, or from the corresponding authors upon request. Source data are published alongside this paper. Source data are provided with this paper.

## Code availability
The tight-binding calculations were performed using a custom-made code on the MATLAB platform. Details of this tight-binding code can be obtained from the corresponding author on request.

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

## Acknowledgements

S. W. acknowledges the financial support from National Key R&D Program of China (No. 2020YFA0309000), the National Natural Science Foundation of China (No. 11874258, No. 12074247), the Shanghai Municipal Science and Technology Qi Ming Xing Project (No. 20QA1405100), Fok Ying Tung Foundation for young researchers and SJTU (No.21×010200846). P.Y. gratefully acknowledges the financial support from the Science and Technology Commission of Shanghai Municipality (20ZR1436900) and ShanghaiTech start-up funding. X.S. acknowledges the Postdoctoral Science Foundation of China (2021M702188). T.F. and R.O. acknowledge support by the Spanish MCIN/AEI/10.13039/501100011033 (PID2020-115406GB-I00), the Basque Department of Education (PIBA-2020-1-0014), and the European Union (EU) through Horizon 2020 (FET-Open project "SPRING" Grant No.863098).

## Author contributions

X.S., Q.D., and P.Y. conceived the experiments. P.Y., S.W., T.F., R.O. supervised the project. Q.D., K.K.Y. synthesized the molecular precursor. X.S. measured the experimental data. Y.L.,Y.J. and C.L., performed the TB calculations. R.O. and T.F. performed CAS-Hubbard and CAS-NEVPT2 calculations. All the authors discussed the results and commented on the manuscript. Q.D., X.S., Y.L., and Y.J. contributed equally to this work.

## Competing interests

The authors declare no competing interests.
