## [Peer Review File · Nature Communications]

Reviewers' Comments:

Reviewer #1:

Remarks to the Author:

This manuscript by Q. Du, et al. reports a comprehensive investigation of the magnetic exchange coupling in a set of nanographene spin dimers and a spin trimer. The authors first used tight-binding calculation to formulate a hypothesis that the symmetry of the frontier orbital have a significant impact on the magnetic coupling. This hypothesis was then subjected to further analysis by DFT and CAS-CI. The on-surface synthesis and SPM experiments demonstrated different coupling strengths for different variations of the trimer molecule on Au(111) surface. Overall the manuscript is interesting but would require major revisions before it could be considered for publication.

Comments:

- a. The AFM result for the D6 molecule, as depicted in Figure 5a, exhibit equal brightness in the right sp³ carbon and the lower region, as opposed to the D1 molecule in the same figure. This observation is insufficient to determine if the sp³ carbon is the only atom passivated with two hydrogens, and thus cannot establish the origin of the Kondo peak at position 1.
- b. The quality of the dI/dV maps Figure 6 e is inadequate to support the conclusions about the orbital and coupling information between positions 1 and 2. Additionally, the labels in Figure 6 are misplaced.
- c. The large range of the dI/dV spectra with Au(111) background is not provided, which is not enough to prove the validity of the results.
- d. The theoretical part of the study primarily relies on qualitative TB-MFH simulations, with the interpretation of the calculation results being largely guided by chemical intuition. We recommend this aspect of the study should get mostly revised with more robust and quantitative methods. Here are specific criticisms of the theoretical part:
 - i. The discussion of the t₃ hopping term is unnecessary. There is no reason to fit the DFT results by considering the t₃ hopping, and then derive a chemical intuition based on only four cases with qualitative results.
 - ii. Furthermore, Figure 3 depicts a case with U = 0 eV, which differs from the simulations of D1-D6 in Figure 1 and 2. It is therefore inaccurate to compare these results, draw conclusions, and make statements.
 - iii. In Figure 3, the explanation for why D1, D2 and D5 have different results with D4 when using the CAS-CI methods is inconsistent with the MFH results is unclear. The reason for larger CAS-CI gaps of is D1, D2, D5 is not well explained, especially with similar structure D4. This means that the MFH results can only be considered as a rough reference, rather than accurate scientific information. Further theoretical work is recommended to address this issue.
 - iv. The analysis of D1-D6 is mainly conducted using MFH and DFT, and there are no corresponding MFH and DFT results for the spin trimer T.
 - v. In Figure 6, the CAS-Hubbard model used to fit the experimental data of the spin trimer T employed U = 4.1 eV, which is not the same as U = |2t| in the cited two paper for this method. As illustrate in the previous part of the paper, different U and t parameter significantly affect the simulation results.
- e. The authors assert in the second sentence of paragraph 4, that the D3 and D6 configurations are well understood by theory and experiments, without providing a reference to support this claim.
- f. The introduction and conclusion mention the potential for room-temperature spintronics, however, all the experiments were conducted at 4K. We suggest either remove this claim or try solution EPR at room temperature to validate the spin interaction.

Reviewer #2:

Remarks to the Author:

The manuscript "Wide Tuning of Magnetic Exchange Coupling in Nanographenes through Orbital-Symmetry Engineering" by Du and his/her colleagues describe the magnetic exchange interaction in the on-surface synthesized (OSS) open-shell nanographenes with a combination of low-temperature scanning spectroscopy/inelastic tunneling spectroscopy and three-different theoretical

calculations (DFT, MFH, and CAS). The authors found two different magnitudes of the exchange coupling, which was attributed to come from the different frontier orbital symmetries. In the last years, open-shell molecules, including triangulates, have attracted interests of researchers because the magnetic coupling can be tuned by OSS precursor molecules in a controlled way. This study can be categorized in this hot field. The topic described in this manuscript would be interesting, yet this reviewer feels that the quality of the work at present does not meet a standard of Nature Communications. Apparently, the theory part leads this study, yet the experimental results does not support the finding accordingly. First of all, the theory considered the dimer of the molecule but no experimental result. It is of importance to show the experimental results with the same system as the one described in the theory. In other word, it is mandatory that the authors measure the magnetic coupling of the compounds described in Figure 1. The theoretical work is based on three-different codes. However, there is no sufficient description why the authors used them. Besides these, all calculation seems to be conducted in gas phase. On surface, there must be a significant interaction between the metal substrate and the radical molecule. It would be very beneficial that the authors include it. Above the reasons, this reviewer feels that the present manuscript is not suitable for the publication. Several minor comments are described as below

1. The title

The authors did not tune the magnetic exchange coupling just gives a few examples of different coupling in the products

2. Fig. 3b

D1, D3, D2, and D5 are shown in this order. Yet, it is a bit confusing because Figure 3a is not.

3. S=1 antiferromagnetic coupling in inelastic tunneling spectroscopy data

The authors obtained the values of 20 mV and 160 mV. How did the authors get them? The AC voltages for the lock in measurement seems to be too large.

4. Page 14, line 230,

"... Consistent with experiments (Fig. 6g)." which is misleading. It is better to change the position of "(Fig. 6)".

5. Page 12, line 199.

By sharp contrast?

6. Linker 1 and Linker 2

It is shown in Figure 1 but is not clear enough in the text.

Response to Reviewer 1:

Comments: This manuscript by Q. Du, et al. reports a comprehensive investigation of the magnetic exchange coupling in a set of nanographene spin dimers and a spin trimer. The authors first used tight-binding calculation to formulate a hypothesis that the symmetry of the frontier orbital have a significant impact on the magnetic coupling. This hypothesis was then subjected to further analysis by DFT and CAS-CI. The on-surface synthesis and SPM experiments demonstrated different coupling strengths for different variations of the trimer molecule on Au(111) surface. Overall the manuscript is interesting but would require major revisions before it could be considered for publication.

We thank the Reviewer for finding our work is interesting. We also appreciate the Reviewer's comments, which we have been fully addressed in the following and modified our manuscript accordingly.

1. The AFM result for the D6 molecule, as depicted in Figure 5a, exhibit equal brightness in the right sp^3 carbon and the lower region, as opposed to the D1 molecule in the same figure. This observation is insufficient to determine if the sp^3 carbon is the only atom passivated with two hydrogens, and thus cannot establish the origin of the Kondo peak at position 1.

We thank the reviewer's comments. The brightness in the lower region of the D6 molecule in Fig.5a is due to the tip creeping towards the sample which leads to more repulsive force detected in the AFM image. If there is additional hydrogen passivated in the lower part, the kondo peak can't be detected at position 1 due to the radical quenching. To avoid any confusion, we have exchanged this AFM image with the one measured in the stabilized tip-sample distance on the same D6 molecule. This AFM image clearly shows that only the sp^3 carbon located at the top-right phenalenyl group exhibits bright contrast due to the additional hydrogen passivation.

Modification: The AFM image of the D6 molecule is exchanged in Fig.5a.

2. The quality of the dI/dV maps Figure 6 e is inadequate to support the conclusions about the orbital and coupling information between positions 1 and 2. Additionally, the labels in Figure 6 are misplaced.

We agree with the Reviewer that the features of dI/dV mappings in previous Fig.6e are not very pronounced. To improve it, we have exchanged these mappings with better quality as below. These dI/dV mappings clearly show that the electronic features at ± 160 mV are mainly located at the top-right and bottom phenalenyl groups, while the electronic states measured at 1 mV are mainly located at the top-left phenalenyl group. These electronic state distributions are consistent with the dI/dV spectra measurements, corresponding to the spin excitation (± 160 mV) and Kondo (1 mV) mappings respectively.

We also thank the Reviewer to point out the error in the figure labels. We have checked all the figure labels, and they are all correct now.

Modification: The dI/dV mappings are replaced with the ones with better quality in Fig.6e. The errors in the Figure captions are all corrected.

3. The large range of the dI/dV spectra with Au(111) background is not provided, which is not enough to prove the validity of the results.

We thank the Reviewer's comment. In the revised manuscript, we have included large range dI/dV spectra of molecules D4, D5, and D6 with Au(111) background in the supporting information. In the cases of molecules D4 and D5, the spectra are measured with the voltages from -1 V to 1.5 V. Because these two molecules have sp^3 hydrogen passivation, the bias voltages are used smaller than 1.5 V to avoid hydrogen dissociation. In this range, the spin-flip feature can also be observed clearly at ± 20 mV and ± 160 mV for D4 and D5 respectively. Moreover, their singly unoccupied/occupied molecular orbital (SUMO/SOMO) can also be detected at 0.45

V/0.95 V and -0.35 V/1.05 V for D4 and D5 respectively. The dI/dV mappings measured at the energies of SOMO/SUMO show similar features as the spin excitation mappings, which can further support the magnetic ground states of the molecules. Considering molecule D6 without any additional hydrogen passivation, the dI/dV spectra can be measured from -2 V to 2 V. The molecular orbitals such as the SOMO/SUMO and the highest occupied/lowest unoccupied molecular orbital (HOMO/LUMO) can be all detected. The dI/dV mappings are also provided at these molecular orbital resonances in the supporting information for reference.

Fig. 1 Large range dI/dV spectra of D4 and D5.

Fig. 2 Large range dI/dV spectra of D6

Modifications: Large range dI/dV spectra with Au(111) background and the corresponding dI/dV mappings of D4, D5 and D6 are included in the supporting information.

4. The theoretical part of the study primarily relies on qualitative TB-MFH simulations, with the interpretation of the calculation results being largely guided by chemical intuition. We recommend this aspect of the study should get mostly revised with more robust and quantitative methods. Here are specific criticisms of the theoretical part:

We thank the Reviewer's comments, which help us to improve our manuscript.

i. The discussion of the t_3 hopping term is unnecessary. There is no reason to fit the DFT results by considering the t_3 hopping, and then derive a chemical intuition based on only four cases with qualitative results.

We apologize that the discussion on the t_3 hopping term is not clear enough in the previous manuscript. Actually, we find that the strengths of exchange interaction have a remarkable difference between D4 and D5 molecules in the experiments first. And then, to understand this large difference, we have calculated the ground and first excited spin states of molecules D4 and D5 with different methods such as MFH, DFT, and CAS-Hubbard. All the calculation results show the same qualitative trend that the D5 molecule has a larger strength of magnetic exchange than that of D4, which is due to the frontier orbital symmetry affecting the singly occupied orbital density distribution. But for the quantitative values, only CAS-Hubbard method employing the t_3 hopping term can best agree with the experimental values. Therefore, we suggest that the t_3 hopping term also has a role in affecting the magnetic exchange coupling. But we agree that it is not necessary to fit the DFT results with the t_3 hopping term. And we need a more robust and quantitative calculation method to compare.

To avoid any confusion and improve the theoretical discussions, we have modified our manuscript structure by presenting the experimental results first and then explaining the mechanism with different calculation methods. In the revised manuscript, we only discuss the molecular structures the same as the experimental results, which have been renamed for phenalenyl diradicals (D1, D2) and triradical (D3). Since DFT and MFH methods are not precise enough to calculate the excitation energies quantitatively due to the spin contamination, we have moved these results to the supporting information only for identifying the ground magnetic states of the molecules. For comparison, we have performed additional calculations by using a robust quantum

chemistry method NEVPT2, which has included the t_3 hopping term spontaneously. Since the calculation results of NEVPT2 and CAS-Hubbard method including the t_3 hopping term can best agree with the experimental values, we suggest that the t_3 hopping term should have a role in affecting the values of exchange interaction.

Modification: 1. We modified the manuscript structure by presenting the experimental results first and then discussed the theoretical mechanism.

2. We included new calculation results by using a robust quantum chemistry method NEVPT2.

3. The calculation results are replaced using the same molecular models as experiments labeled as D1, D2 and D3 for phenalenyl diradicals and triradical.

4. We modified the discussion on the effect of the t_3 hopping term at page 12.

“As shown in Fig.4b, although the calculated exchange interaction of **D1** is smaller than that of **D2** by CAS-Hubbard only considering the t_1 hopping term, the exchange interaction of **D1** and **D2** obtained by this method is remarkably different from the experimental values. To quantitatively explain the experiments, we find that upon employing the t_3 hopping term in the CAS-Hubbard calculation, the calculated exchange interaction of **D1** decreases, while increases for **D2**, both agreeing better with the experimental values. These results suggest that the t_3 hopping term also has an impact on the exchange interaction, that is already included in the NEVPT2 calculation. (The next-nearest hopping t_2 of graphene honeycomb lattice is less important in our case, cf. supplementary information).”

ii. Furthermore, Figure 3 depicts a case with $U = 0$ eV, which differs from the simulations of D1-D6 in Figure 1 and 2. It is therefore inaccurate to compare these results, draw conclusions, and make statements.

We thank the Reviewer's comments. In Figure 3, the calculation is performed without considering the electron-electron interaction ($U=0$), just to easily compare the wave function symmetry on the molecular conjugation structures. If we include U values larger than 3.5 eV in the calculation, the molecules' ground state will turn from the closed shell to the open shell. And the linear combination of the HOMO and LUMO wavefunctions with $U=0$ can obtain the singly occupied and unoccupied molecular orbitals (SOMOs/SUMOs) in the case of $U = 3.5$ eV. We have added the low-lying orbitals with $U=3.5$ eV for comparison. It is shown that the SOMOs are exactly the same as the constructed orbitals using HOMO and LUMO with $U=0$.

As the results shown in below, both for $U=0$ and $U=3.5$, the constructed wave function of SOMOs is likely to propagate in the branch of the molecular backbone along the

mirror plane direction in D1, while the SOMOs wave function does not propagate along the mirror plane direction in D2. Since the magnetic exchange interaction depends on the orbital overlap of SOMOs at the connecting region (near the mirror plane), the D1 configuration has a much larger exchange coupling strength than D2 due to large orbital density. We have included results with $U = 3.5$ eV in the supporting information.

Modification: We included the results of wave function distribution with $U = 0$ eV and $U = 3.5$ eV for comparison in the supporting information.

iii. In Figure 3, the explanation for why D1, D2 and D5 have different results with D4 when using the CAS-CI methods is inconsistent with the MFH results is unclear. The reason for larger CAS-CI gaps of D1, D2, D5 is not well explained, especially with similar structure D4. This means that the MFH results can only be considered as a rough reference, rather than accurate scientific information. Further theoretical work is recommended to address this issue.

We agree with the Reviewer that MFH results can only be considered as a rough reference since DFT and MFH can't quantitatively calculate the excitation energy due to the spin contamination. In the revised manuscript, we have moved these results to the supporting information only for identifying the magnetic ground state. Moreover, a robust quantum chemistry method NEVPT2 was employed in the calculation. All the molecular models used for theoretical calculations are exchanged for complying with the experimental results, which have been labeled as D1 and D2 for two kinds of phenalenyl diradicals.

Considering the experimental result that the spin excitation voltage of D1 (20 meV) is much smaller than that of D2 (160 meV), it mainly originated from the molecular orbital symmetry affecting the singly occupied orbital density distribution, which determines the strength of magnetic exchange interaction. In the case of D1, its HOMO wave function has an antisymmetric character exhibiting the opposite signs of wave functions corresponding to the middle mirror plane normal to the molecule backbone. And this antisymmetric wave function prefers to propagate along the mirror plane direction. Therefore, the singly occupied orbital density per site at the connection region is effectively reduced, which leads to a smaller strength of exchange interaction of D1. In contrast, D2 has symmetric wave functions of HOMO, which does not prefer to propagate in the mirror plane direction normal to the molecular backbone. Therefore, its singly occupied orbital density distribution is not reduced at the connection region. That's why the strength of the magnetic exchange of D1 is much smaller than that of D2.

Modification: 1. The MFH calculation results are moved to the supporting information.
2. Additional theoretical works using NEVPT2 is employed.
3. Molecular models used for calculations are exchanged for complying with the experimental results.
4. The discussion on the reason of exchange interaction difference between D1 and D2 is modified at page 11 and 12.

“As highlighted in Fig. 5b, for the molecular **D1**, the wave function of its HOMO has antisymmetric character showing the opposite sign of the wave function to the mirror plane perpendicular to the molecular backbone. This antisymmetric wave function is likely to propagate in the branch of the molecular backbone along the mirror plane direction. Thus, the singly occupied orbital density per site at the connecting region of **D1** is effectively reduced, resulting in a smaller exchange interaction strength for **D1**. In contrast, **D2** has a symmetric HOMO, whose wave function does not propagate in the branch of the molecular backbone along the mirror plane direction. Therefore, the singly occupied orbital density distribution is not reduced at the connecting region for **D2**. As a consequence, **D1** has a smaller coupling strength than **D2** due to the reduced orbital density overlap at the connecting region, originated from frontier orbital symmetry differences.”

iv. The analysis of D1-D6 is mainly conducted using MFH and DFT, and there are no corresponding MFH and DFT results for the spin trimer T.

In the case of spin triradical T, there are three coupled spins. Because of the neglected spin-orbital coupling in the graphene system, the system can be simplified as a three-site Heisenberg quantum spin model. In the picture of mean-field approximation, the wave functions are not the eigenstates of the total spin operator \hat{S} , but of the \hat{S}_z projection. So mean-field DFT and MFH calculations are not reliable for excitation energies calculation for multiple spin systems. We have performed DFT and MFH for the spin triradical T and found the obtained spin states greatly deviate from experiments.

v. In Figure 6, the CAS-Hubbard model used to fit the experimental data of the spin trimer T employed $U = 4.1$ eV, which is not the same as $U = |2t|$ in the cited two paper for this method. As illustrate in the previous part of the paper, different U and t parameter significantly affect the simulation results.

We thank the reviewer’s comments. The Hubbard model is used to qualitatively address the low-lying states of the system. We found the values of the chosen U inside a certain range (usually $t < U < 2.2t$) give the same magnetic ground states. Therefore, it is not necessary to use exactly the same values as the cited two papers for this method.

5. The authors assert in the second sentence of paragraph 4, that the D3 and D6

configurations are well understood by theory and experiments, without providing a reference to support this claim.

We thank the Reviewer to point this out. D3 and D6 molecules are two conjugated phenalenyl groups in the way of hosting sublattice imbalance as suggested in Fig1. Therefore, they have the high spin ground state of $S=1$. The nanographene with a high spin ground state due to the sublattice imbalance has been investigated by several theoretical and experimental works. To make this point clear, we have modified this sentence and cited relative works on it.

Modification: we modified the sentence on the magnetic state of D3 and added related references at page 9 and 10.

“Since **D3** hosting the high spin ground state, due to the sublattice imbalance, is well understood by theory²⁵ and experiments^{41,42}, we focus on the competing cases of **D1** and **D2** as depicted in Fig. 4.”

6. The introduction and conclusion mention the potential for room-temperature spintronics, however, all the experiments were conducted at 4K. We suggest either remove this claim or try solution EPR at room temperature to validate the spin interaction.

We thank the reviewer’s comment. Since one of our investigated molecules presents a large exchange interaction, which significantly exceeds the room-temperature thermal energy (25 meV), we suggest it has the potential to have applications at room temperature. Considering the reviewer’s suggestion, we don’t emphasize this point in the revised manuscript and removed the words “room-temperature”.

Modifications: We removed the words “room-temperature” in the introduction and conclusion parts.

Response to Reviewer 2:

Comments: The manuscript “Wide Tuning of Magnetic Exchange Coupling in Nanographenes through Orbital-Symmetry Engineering” by Du and his/her colleagues describe the magnetic exchange interaction in the on-surface synthesized (OSS) open-shell nanographenes with a combination of low-temperature scanning spectroscopy/inelastic tunneling spectroscopy and three-different theoretical calculations (DFT, MFH, and CAS). The authors found two different magnitudes of the exchange coupling, which was attributed to come from the different frontier orbital

symmetries. In the last years, open-shell molecules, including triangulates, have attracted interests of researchers because the magnetic coupling can be tuned by OSS precursor molecules in a controlled way. This study can be categorized in this hot field. The topic described in this manuscript would be interesting, yet this reviewer feels that the quality of the work at present does not meet a standard of Nature Communications.

We thank the Reviewer finds the topic of our work is interesting. We have modified our manuscript according to both the reviewers' comments. We hope our revised manuscript can meet the standard of Nature Communications.

Apparently, the theory part leads this study, yet the experimental results does not support the finding accordingly. First of all, the theory considered the dimer of the molecule but no experimental result. It is of importance to show the experimental results with the same system as the one described in the theory. In other word, it is mandatory that the authors measure the magnetic coupling of the compounds described in Figure 1.

We apologize that we have not presented our work clearly in our previous manuscript. In fact, we first find interesting experimental evidence that just by tuning a little bit the conjugation symmetry, the strength of magnetic exchange interaction can be varied about one order of magnitude. To explain this experimental finding, we employed several calculation methods to find out the physical mechanism.

To avoid any confusion, we have modified the structure of our manuscript. We first present the experimental results and then demonstrate the calculation results. Moreover, we agree with the reviewer that the molecular models used for calculation should be consistent with the experimental results. In the revised manuscript, all the calculation results presented are calculated with the molecular models complying with the experimental results.

Modifications: 1. We modified the manuscript structure by presenting the experimental results first and then explained the mechanism with different calculation methods.

2. The calculation results are presented with the same molecular models as the experiments.

The theoretical work is based on three-different codes. However, there is no sufficient description why the authors used them.

We thank the Reviewer's comments. DFT and MFH calculations are based on mean-field approximations, which normally underestimate the coulomb interactions. So we

used high-level CAS-Hubbard and NEVPT2 calculations to better capture the many-body interactions.

We realized that in our previous manuscript too many calculation models are presented, which may confuse the readers. Therefore, these calculation results are moved to the supporting information only for identifying the magnetic ground states in the revised manuscript. Now, we only use CAS-Hubbard and a more robust quantum chemistry method NEVPT2 for comparison. The corresponding discussions are also modified.

Modifications: 1. The MFH and DFT calculation results are moved to the supporting information.

2. Additional theoretical works using NEVPT2 are employed.

3. The corresponding discussion is modified.

Besides these, all calculation seems to be conducted in gas phase. On surface, there must be a significant interaction between the metal substrate and the radical molecule. It would be very beneficial that the authors include it.

We thank the Reviewer's comments. Previous DFT calculations suggest the Au(111) substrate doesn't affect the magnetic ground states of magnetic triangulene systems, where the gas phase calculations agree well with experimental results (Physical Review B 104, 075404 (2021)). Considering our large system, a full quantum chemistry calculation including the substrate is too heavy to be realized.

Above the reasons, this reviewer feels that the present manuscript is not suitable for the publication. Several minor comments are described as below

With our responses and the corresponding modifications, we hope our revised manuscript can meet the criterion of publication in Nature Communications. For the other comments, we responded as below.

1. The title

The authors did not tune the magnetic exchange coupling just gives a few examples of different coupling in the products

We agree that in the experiments we only observe two kinds of nanographenes with huge different strengths of exchange interaction determined by their wave function symmetries. To precisely present our work, we modified our manuscript title to "Orbital-Symmetry Effects on Magnetic Exchange in Open-Shell Nanographenes".

Modification: The title is modified.

2.Fig. 3b

D1, D3, D2, and D5 are shown in this order. Yet, it is a bit confusing because Figure 3a is not.

We agree with the reviewer that we have shown too many molecular models in Fig.1, which makes it difficult to understand our main experimental and theoretical results. To avoid any confusion, we have modified molecular models in Fig.1, which are consistent with the experimental results. In the revised manuscript, we only discuss two nanographene diradicals (D1, D2) and one nanographene triradical (D3), which have been investigated in the experiments. It should give a better understanding for both the experimental and theoretical results.

Modifications: 1. Figure. 1 is modified.

2. Only the molecular structures observed in the experiments are discussed in the revised manuscript, which have been labeled as D1, D2 and D3.

3. S=1 antiferromagnetic coupling in inelastic tunneling spectroscopy data

The authors obtained the values of 20 mV and 160 mV. How did the authors get them? The AC voltages for the lock in measurement seems to be too large.

The values of 20 meV and 160 meV are obtained by measuring inelastic tunneling spectra. As shown below, the spin excitation threshold energies are about ± 20 mV and ± 160 mV at the step position in the dI/dV spectra for D1 and D2 respectively. To determine the excitation voltage, the second dI/dV spectra can also be used (the figures in the second row). The peak positions in the second dI/dV spectra correspond to the step position in the dI/dV spectra, which both give the spin excitation values of ± 20 mV and ± 160 mV for D1 and D2 respectively. We have included the second dI/dV spectra in the supporting information for reference.

Fig. 3 The dI/dV (top) and second dI/dV (bottom) spectra of D1 and D2.

For the inelastic tunneling spectra measurements, we have tried different modulation voltages. As shown below, using 5 mV or 1 mV of modulation voltage both give similar spin-flip spectra. In the previous manuscript, 20 mV is a typo error for the excitation mapping measurement condition. This modulation voltage should also be 5 mV, which has been corrected in the revised manuscript.

Fig. 4 The dI/dV spectra measured with different lock-in modulation voltages.

Modifications: 1. The typo error is corrected.

2. The second dI/dV spectra are included in the supporting information.

4. Page 14, line 230,

“... Consistent with experiments (Fig. 6g).” which is misleading. It is better to change the position of “(Fig. 6)”.

We thank the Reviewer’s suggestion. We have modified this sentence.

Modifications: This sentence at page 14 is modified.

“To understand this system, a Heisenberg spin trimer model (Fig.6f) is solved, the calculation results shown in Fig.6g demonstrate that the ground state has $S = 1/2$ and the other six excited states are nearly degenerate with an excitation gap of ~ 160 meV.”

5. Page 12, line 199.

By sharp contrast?

We would like to emphasize the remarkable difference in magnetic exchange between molecules D1 and D2, which is about one order of magnitude. To avoid any confusion, this sentence has been modified.

Modifications: The sentence at page 8 is modified.

“In sharp contrast with the magnetic exchange of 20 meV observed on D1, the corresponding dI/dV spectra taken on D2 exhibit a substantially increased excitation threshold of 160 meV.”

6. Linker 1 and Linker 2

It is shown in Figure 1 but is not clear enough in the text.

We thank the Reviewer’s comment. We have realized that there are too many molecular models in the previous manuscript, which are difficult for the readers to catch the main experimental and theoretical results. In the revised manuscript, we have modified our molecular models for consistent with the experimental results and there is no need to introduce linker 1 and linker 2.

Modification: We have modified Fig.1, the linkers 1 and 2 have been removed.

Reviewers' Comments:

Reviewer #1:

Remarks to the Author:

The manuscript has been significantly improved and the authors have addressed almost all the points raised as part of the review. In its current form the manuscript could be considered for publication in Nature Communications.

Reviewer #2:

Remarks to the Author:

The authors changed the structure of the manuscript as showing the experimental data and subsequently explaining the possible mechanism with the theoretical calculation. Now, I think that the readability of the manuscript improves. However, this reviewer found it difficult to read the response letter. For instance, the authors states that "we have included large range dI/dV spectra of molecules D4, D5, and D6..." but no related data was shown in supporting information. The authors also replied as ", which have been renamed for phenalenyl diradical (D1, D2) and triradical (D3)" but D3 seems to be a diradical. Most probably all would relate to typos but this reviewer is afraid that the intentions of the authors are really understood correctly.

The new dI/dV data in Supplementary Figure 16 shows distinct different peak heights in second differential curves for D2 but not for D1. Why? Also, this reviewer could not find the zero line in the d²I/dV² curves (btw, the authors described "d²I/dV²", which is incorrect.). The contrast of SOMO for D1 is almost identical with that of SUMO, but D2 shows distinct contrasts (Supplementary Figure 17). Why? These points relate to Comment 2 of Reviewer 1. The authors just replaced the dI/dV maps taken at +/- 160 mV without any adequate explanation. For these reasons, this reviewer still feels that this contribution does not meet the standard for publication in Nature Communication.

Minor:

1. Chemical structures in Figure 4 should be revised.
2. Bond-resolved AFM. Is that the topography? No information can be found in the figure.

Response to Reviewer 1:

Comments: The manuscript has been significantly improved and the authors have addressed almost all the points raised as part of the review. In its current form the manuscript could be considered for publication in Nature Communications.

We thank the Reviewer for his/her positive evaluation of our work.

Response to Reviewer 2:

Comments: The authors changed the structure of the manuscript as showing the experimental data and subsequently explaining the possible mechanism with the theoretical calculation. Now, I think that the readability of the manuscript improves.

We thank the Reviewer finds our revised manuscript has improved.

However, this reviewer found it difficult to read the response letter. For instance, the authors states that “we have included large range dI/dV spectra of molecules D4, D5, and D6...” but no related data was shown in supporting information.

These data have been included in the supporting information in Supplementary Figure 17 and Supplementary Figure 18 (p.14 and p.15 in the supporting information), but with the revised labels as D1, D2, and T instead of D4, D5, and D6.

The authors also replied as “, which have been renamed for phenalenyl diradical (D1, D2) and triradical (D3)” but D3 seems to be a diradical. Most probably all would relate to typos but this reviewer is afraid that the intentions of the authors are really understood correctly.

We apologize that this is the typo error in the previous response letter. As the reviewer’s understanding, D1, D2, and D3 are phenaleny diradicals, while phenaleny triradical is labeled as T. We have checked through our revised manuscript, these labels should be correct and consistent.

The new dI/dV data in Supplementary Figure 16 shows distinct different peak heights in second differential curves for D2 but not for D1. Why?

We thank the Reviewer’s comments. The bias-asymmetric peak heights are commonly observed in the experiments. The reason has been systematically investigated in theory (e.g. New J. Phys. 17, 63016 (2015)), which suggests that the high-order

scattering process will result in the asymmetric peak heights in the IETS spectrum. And particularly, if there is a resonant state near the bias window of spin excitation, the high-order scattering process will be enhanced and produce a more pronounced asymmetric IETS line shape. As shown in the following figure, this is the case that we observed in our d^2I/dV^2 . Since the spin excitation energies are very small for D1 (± 20 mV), which are both far away from its resonant orbital states of SOMO (-450 mV) and SUMO (950 mV), the asymmetric peak heights of D1 d^2I/dV^2 spectrum is not obvious. In contrast, the spin excitation energies of D2 are much larger (± 160 mV), which is close to its SOMO (~ 350 mV) resonant orbital in the negative bias direction but far away from its SUMO orbital (1.05 V). Therefore, the peak height is more pronounced in the negative bias and results in a huge asymmetric IETS spectrum. This large asymmetric peak height behavior has also been observed recently in the d^2I/dV^2 spectrum of a nanographene system with large exchange interaction, whose spin excitation energy is also close to one of the molecular resonant states (J. Am. Chem. Soc. 145, 2968-2974 (2023)). To clarify this point, we have added some discussion on this asymmetric line shape in the manuscript.

Modification: We added some discussion at page 8 and cited related theoretical and experimental works.

“And a more pronounced asymmetric line shape of dI/dV spectra is observed on D2 compared to that of D1 since D2 has a high spin excitation energy which is close to its SOMO resonant state thus leading to a pronounced height in the negative bias.^{32,57}”

Also, this reviewer could not find the zero line in the d^2I/dV^2 curves (btw, the authors described “ d^2I/dV^2 ”, which is incorrect.).

We thank the reviewer’s careful check. In the previous version, we shifted the spectra measured on the molecule and on Au(111) substrate vertically for clarity without

indicating the zero line. In the revised version, we canceled the vertical shift (cf. the following figure).

Modification: The vertical ordinates are added in the d^2I/dV^2 spectra in Supplementary Fig.16. The typo error of d^2I/dV^2 is also corrected.

The contrast of SOMO for D1 is almost identical with that of SUMO, but D2 shows distinct contrasts (Supplementary Figure 17). Why?

The SUMO contrast of D2 is distinct from its SOMO mainly due to their different tip-molecule separations when measuring the SOMO and SUMO dI/dV mappings. Since D1 and D2 have sp^3 hydrogen passivation, the additional hydrogen will be dissociated if the tip is too close to the sample. And because the SUMO energy of D2 is about 1.05 eV while its SOMO energy is only 0.35 eV, the mappings of SUMO and SOMO are taken at different tunneling current set points of $I = 400$ and 500 pA respectively to avoid hydrogen dissociation. Therefore, the tip-molecule separation at measuring the dI/dV mappings is much larger for SUMO than that of SOMO, which results in the fine structure of SUMO being different from SOMO. Additionally, we performed simulations considering such different tip-sample separation effect. As seen in the following simulated STS mappings, the fine structures of SOMO/SUMO orbitals depend sensitively on tip-sample separation, consistent with experimental observations.

Modification: The simulated dI/dV mappings of SOMO and SUMO at different tip-molecule distance have been included in the Supplementary Figure 17.

These points relate to Comment 2 of Reviewer 1. The authors just replaced the dI/dV maps taken at ± 160 mV without any adequate explanation.

To detect the fine structures of molecular orbitals, we usually use CO functionalized tip since the p-wave orbitals of the CO molecule can enhance the lateral resolution compared with a bare metal tip (Phys. Rev.Lett. 107, 086101 (2011)). In experiments, the CO molecule is usually picked up to the tip by applying a short pulse, and its adsorption on the tip position sometimes is not perfect vertical and symmetric, which may not give the best spatial resolution as expected. The dI/dV mappings taken at ± 160 mV in the revised manuscript is measured by a CO functionalized tip with a much higher spatial resolution of the orbital distribution than the previous images.

For these reasons, this reviewer still feels that this contribution does not meet the standard for publication in Nature Communication.

With our responses and the corresponding modifications, we hope our revised manuscript can meet the criterion of publication in Nature Communications. For the other comments, we responded as below.

Minor:

1. Chemical structures in Figure 4 should be revised.

We checked the chemical structures in Fig.4 and thought the reviewer might mean the chemical structure is not used the templates of ChemDraw. Therefore, we modified the chemical structure by using the ChemDraw template in Fig.4

Modification: The chemical structure in Fig.4 is modified using the ChemDraw template.

2. Bond-resolved AFM. Is that the topography? No information can be found in the figure.

The bond-resolved AFM is measured by non-contact atomic force microscope with a qPlus sensor. The AFM images were recorded with the different shift frequency of the qPlus sensor in the constant height mode, whose contrasts reflect the topography of the molecules (*Science* 325,1110-1114 (2009)). To make this point more clear, we have added grey scale bars of the shift frequency in all AFM images.

Modification: The AFM images in Fig.2, Fig3, and Fig6 are added with the grey scale bars of the shift frequency.

Reviewers' Comments:

Reviewer #2:

Remarks to the Author:

The authors improved the manuscript a lot. Now, this reviewer feels more confident that this contribution deserves publication in Nature Communications.